# OpenReview forum: "A Concept is More Than a Word: Diversified Unlearning in Text-to-Image Diffusion Models"
_ICLR.cc/2026/Conference — Submitted to ICLR 2026_

### Official Review · Reviewer_jidj · 2025-10-20

**Soundness:** 2
**Presentation:** 3
**Contribution:** 2
**Rating:** 2
**Confidence:** 3

**Summary:**

This paper proposes a Diversified Unlearning approach to enhance the robustness for concept unlearning tasks. The proposed method adds a set of contextually diverse prompts to augment the original keyword. This mechanism is realized by prompting ChatGPT to generate the context set. The experiment results demonstrate that the proposed method can achieve effective concept unlearning and be robust to adversarial attack prompts.

**Strengths:**

1. Improving the robustness of concept unlearning methods is crucial and practical for real-world trustworthy generative AI developments, especially for malicious attack prompts.
2. The overall paper is easy to follow.

**Weaknesses:**

1. In this paper, the contextualized prompts c are generated by ChatGPT. However, how to decide the number of contextualized prompts that need to be generated for each task?
2. In addition, LLMs inevitably have hallucination issues. How to make sure the generated contextualized diverse prompts are reasonable and faithful, so that they can properly describe the variations of the target concept?
3. Previous works, such as Receler (ECCV’24), adopt learnable prompt embeddings to handle the rephrased concepts by adversarial training. They also emphasize the focus on enhancing robustness. Comparing the contextualized prompts from ChatGPT with the learnable prompts would make the major claimed technical contributions more convincing.
4. In lines 43-44, this paper claims that the proposed method is able to preserve the unrelated concepts. However, from Eq.3 and Eq.5, how to construct the unrelated set? It's impractical to cover every unrelated concept in the diffusion models. It would be beneficial to provide clearer clarifications and details in the paper.

**Questions:**

1. How to ensure the output images contain safe concepts? It is possible that while the target concept is removed, it generates other unsafe concepts.

---

> ### Author Response · Authors · 2025-11-22
> **Author's responses**
>
> We thank the review for the positive feedback on the improvement made by our method. We would like to address the remaining concerns as follows:
>
> **Q1. How do you ensure that the output images contain safe concepts? Is it possible that, while the target concept is removed, the model generates other unsafe concepts?**
>
> In our experiments, we follow—and extend—the standard evaluation protocol commonly used in the machine unlearning literature. As summarized in Table 4, we evaluate unlearning across a range of concepts, including celebrities, objects, copyrighted characters, artistic styles, and nudity. To assess unlearning performance, prompts are designed to *generate* the target concept. This means that if the target concept itself is safe (e.g., a celebrity or an object), we do not use prompts containing unsafe concepts, consistent with established practice in prior work.
>
> For the nudity category specifically, we adopt the I2P prompt set used in the ESD paper. Additionally, to further assess the robustness of our approach, we employ Ring-A-Bell—a recovery jailbreak attack capable of eliciting nudity-related outputs without explicitly using the keyword *'nudity'*. To determine whether nudity appears in the generated images, we use NudeNet, a pretrained object detection model that reliably identifies inappropriate body parts.
>
> In summary, our experimental setup is aligned with standard practice and includes additional robustness checks to ensure comprehensive evaluation of safety-related behavior. We believe these measures sufficiently validate the effectiveness and safety of our proposed method. We hope this addresses your question.

---

> ### Author Response · Authors · 2025-11-22
> **Author's responses**
>
> **Weakness 1. How to decide the number of contextualized prompts that need to be generated for each task?**
>
> We conducted ablation studies on the number and complexity levels of prompts for celebrity unlearning and discussed the results in Section 4.5. We used these findings to define generalizable rules for scaling across different concepts. As shown in Fig.4b, when using Level-1 prompts, the model maintains strong erasure and preservation performance even as the number of prompts varies from 5 to 50. This indicates that once an appropriate prompt complexity level is chosen, **the quantity of prompts is not the dominant factor affecting performance**. Instead, the core performance gain comes from the design of our diversified prompts rather than from increasing the prompt count. For simplicity, we used 20 contextualized prompts for all experiments, except for the Diversified-UCE which used 5 prompt sets following its closed-form solution.
>
> **Weakness 2. In addition, LLMs inevitably have hallucination issues. How to make sure the generated contextualized diverse prompts are reasonable and faithful, so that they can properly describe the variations of the target concept?**
>
> LLMs hallucination is not a primary concern of this work. While an LLM may occasionally generate unrealistic prompts such as *'Henry Cavill riding a horse on the moon'*, the key question for our setting is whether the *identity* of *'Henry Cavill'* appears in the generated output. Since Diversified Prompting is applied to output-based unlearning methods such as ESD, the output signal is what ultimately matters.
>
> To ensure prompt reliability, we construct a verification pipeline. For each candidate prompt, we generate multiple images using different random seeds and evaluate them with a recognition model aligned with the one used during evaluation. Only prompts whose generated images consistently contain the target concept with high confidence are retained for unlearning.
>
> **Weakness 3. Additional experiment with Receler (ECCV'24)**
>
> We follow the reviewer's suggestion to conduct an additional experiment with Receler (erasing the celebrity *'Henry Cavill'*) and provide the results in the table below.
>
> | **Method**| **Erasure** || **Preservation** ||||
> |-|-|-|-|-|-|-|
> || CLIP-i↓    | GPT-score↓ | LPIPS↓| CLIP-i↑ | CLIP-t↑ | GPT-score↑       |
> | Receler| **51.61**| **0.00**| 0.57| 73.93   | 25.27  | 47.07           |
> | **Diversified-Receler**| 52.90| **0.00**| **0.52**| **77.65** | **26.17** | **55.60** (+8.53%) |
>
> As shown in the table above, integrating Diversified Unlearning into Receler preserves the original model’s erasure performance while markedly enhancing preservation, yielding an 8.53\% improvement in the GPT-score. Beyond the quantitative metrics, we also provide qualitative examples in the anonymous GitHub repository to enable direct visual comparison between Receler and Diversified-Receler. Although additional fine-tuning of our add-on module could potentially lead to even stronger performance, we intentionally employ the same default configuration used in our other experiments to ensure fairness and consistency.
>
> These additional results—together with the comprehensive experiments conducted on recent state-of-the-art unlearning methods such as *ACE} (CVPR 2025)* and *AGE* (ICLR 2025)—further reinforce both the effectiveness and the broad applicability of our proposed approach

---

> ### Author Response · Authors · 2025-11-22
> **Author's responses**
>
> **Weakness 4. From Eq.3 and Eq.5, how to construct the unrelated set? It's impractical to cover every unrelated concept in the diffusion models. It would be beneficial to provide clearer clarifications and details in the paper**
>
> A typical unlearning objective consists of two components: an erasure loss and a preservation loss. In our Diversified Prompting framework for output-based methods, we intervene only in the erasure component while keeping the preservation component identical to that of the baselines. Consequently, we do not introduce any additional mechanisms specifically targeting unrelated concepts. Instead, **the improved preservation of unrelated concepts emerges naturally from the conflict introduced in our redesigned erasure loss, which forces the model to remove only what is necessary**.
>
> Intuitively, keyword-based approaches rely on simple prompts or keywords such as *'Henry Cavill* or *'nudity'* to identify the concept to be erased. However, due to the inherent entanglement among visual concepts, such keywords rarely correspond to a single, isolated concept. For example, as illustrated in Figure 1, the prompt *'a photo of Henry Cavill'* may lead to images depicting his iconic Superman character or scenes involving a child. Similarly, the keyword *'nudity'* may evoke a specific stereotypical depiction (e.g., a photo of a woman without clothes). These unintended associations are often suppressed together with the target concept, leading to reduced preservation performance for keyword-based unlearning methods.
>
> In contrast, our diversified strategy offers a simple yet effective means of disentangling the target concept from its unintended associated concepts by embedding it within diverse contextualizations. Specifically, we erase the target concept by mapping **{'context' + 'target concept'}** to **{'context' + 'neutral anchor concept'}**. This formulation explicitly instructs the model to remove only the target concept while retaining the surrounding context. By enforcing contextual preservation directly within the erasure loss, our method provides clearer and more precise guidance regarding what should and should not be removed, substantially reducing unintended interference with unrelated concepts.

---

> > ### Author Response · Authors · 2025-11-27
> >
> > Dear Reviewer **jidj**,
> >
> > We would like to provide additional experiments on the suggested adversarial-training–based unlearning methods, including **AdvUnlearn**, **RECE**, and **Receler** (previously provided, but reproduced here for clarity). AdvUnlearn and Receler belong to the *output-based unlearning* family, following the ESD framework, while RECE is an *attention-based* method with a closed-form solution similar to UCE.
> >
> > We integrate our proposed diversification technique into these keyword-based frameworks, resulting in three diversified variants: **Diversified-AdvUnlearn**, **Diversified-RECE**, and **Diversified-Receler**. We evaluate them on two erasure tasks: removing the celebrity *Henry Cavill* and removing *ImageNette* objects, using the same experimental settings detailed in Appendix B.1.1. The code has also been updated in the anonymous repository for further inspection.
> >
> > From the first table, we observe that:
> >
> > 1. **Compared to Receler**, our diversified counterpart achieves *equally strong unlearning performance* (both methods reach zero GPT-score and very low CLIP-i), while achieving *substantially better preservation performance*, improving by **+8.53 GPT-score** or **+3.7 CLIP-i**.
> > 2. **Compared to RECE**, our diversified variant achieves slightly better preservation performance, along with a noticeably improved unlearning performance.
> >
> > The second table shows that, relative to AdvUnlearn, our Diversified-AdvUnlearn variant achieves *comparable unlearning effectiveness* while providing a *much stronger preservation ability*, including a notable **+30% PSR-5 improvement** on the task of erasing five objects simultaneously.
> >
> > Taken together, these results—along with the extensive experiments on recent state-of-the-art methods such as **ACE (CVPR 2025)** and **AGE (ICLR 2025)**—further strengthen our claim regarding both the *effectiveness* and the *generality* of our diversified unlearning approach. Our method not only enhances unlearning quality but also consistently improves preservation across diverse keyword-based unlearning frameworks.
> >
> > | **Method**| **Erasure** || **Preservation** ||||
> > |-|-|-|-|-|-|-|
> > || CLIP-i↓    | GPT-score↓ | LPIPS↓| CLIP-i↑ | CLIP-t↑ | GPT-score↑|
> > | **Receler**| **51.61**| **0.00**| 0.57| 73.93   | 25.27  | 47.07|
> > | **Diversified-Receler**| 52.90| **0.00**| **0.52**| **77.65** | **26.17** | **55.60** (+8.53%) |
> > | **RECE**| 59.42| 8.05| **0.79**| 77.34| 29.32| 77.42|
> > | **Diversified-RECE**| **58.57**| **7.78**| **0.79**| **77.95**| **29.64**| **77.87**|
> >
> > Quantitative results of erasing `Henry Cavill' using baseline Receler and RECE compared with the diversified counterpart, as detailed in Appendix B.1.1.
> >
> > | **Object**| **'Garbage Truck' erasure**|||| **'Five objects' erasure**||||
> > |-|-|-|-|-|-|-|-|-|
> > || ESR-1| ESR-5|PSR-1|PSR-5|ESR-1|ESR-5|PSR-1|PSR-5|
> > | **AdvUnlearn**|**1.000**|**0.992**|0.768|0.936|**1.000**|**0.990**|0.459|0.604|
> > | **Diversified-AdvUnlearn**|0.996|0.976|**0.807**|**0.952**|0.988|0.974|**0.732**| **0.916** |
> >
> > Quantitative results of object erasure using keyword-based methods compared with their diversified counterparts, evaluated across prompts of varying complexity levels. Erasure Success Rate (ESR) and Preservation Success Rate (PSR), computed with a pre-trained ResNet-50, where higher values indicate better performance. Overall, under the setting of simultaneously erasing five objects.

---

> ### Author Response · Authors · 2025-11-27
>
> Dear Reviewer **jidj**,
>
> Thank you once again for the time and care you invested in reviewing our submission. We sincerely appreciate your constructive feedback, which has helped us further strengthen the paper. We are writing to kindly remind you that we have now posted a detailed response to your comments.
>
> In our rebuttal, we have specifically addressed the key concerns you raised:
>
> #### **1. Enhancing adversarial training–based unlearning methods**
> Our method functions as an **add-on module** that can be applied on top of existing unlearning baselines. When integrated with adversarial training–based methods such as **Receler**, our approach improves preservation performance while retaining the strong erasure capabilities inherent to these adversarial baselines. Similar improvements are also observed in our new experiments with **AdvUnlearn** and **RECE**.
>
> #### **2. Clarifying the operational mechanism of Diversified Prompting**
> A typical unlearning method comprises two components: an erasure loss and a preservation loss. Diversified Prompting for output-based methods modifies only the **erasure loss**, while keeping the preservation loss from the baseline unchanged. As such, we do not introduce additional mechanisms specifically for “unrelated concepts.”
> Instead, we introduce a form of **contrastive tension** directly within the erasure loss. By crafting prompts with carefully controlled semantic complexity, we embed the target concept into diverse, concrete contexts. This enriches the representational space of the concept to be removed while simultaneously revealing the elements that should be preserved. Consequently, Diversified Prompting achieves more accurate erasure without compromising preservation performance.
>
> #### **3. Prompt design considerations in Diversified Prompting**
> In Section 4.5 and Appendix C.6, we provide an ablation study analyzing how contextual diversity and semantic complexity affect method performance, highlighting three key findings:
> (i) diversity is essential,
> (ii) moderate contextual complexity is most effective, and
> (iii) increasing the number of prompts yields diminishing returns.
>
> Furthermore, hallucination is not a primary factor influencing the performance of our method. What matters more is whether the generated output reliably contains the target concept, as Diversified Prompting is specifically designed for output-based unlearning methods where the presence of the target concept in generated samples is crucial.
>
> ---
>
> We would be sincerely grateful if you could take a moment to review our detailed responses. If you find that our new experiments and clarifications adequately address your concerns, we would deeply appreciate it if you could reconsider your evaluation and score accordingly.
>
> Please feel free to reach out with any further questions during the remaining days of the discussion period — we remain happy to provide any additional clarification.
>
> Thank you again for your valuable insights.
>
> **Best regards,**
> *The Authors of Submission 3912*

---

### Official Review · Reviewer_WWWB · 2025-10-26

**Soundness:** 2
**Presentation:** 3
**Contribution:** 3
**Rating:** 6
**Confidence:** 4

**Summary:**

This paper introduces Diversified Unlearning, a simple text-to-image unlearning framework that enhances the contextual diversity of textual prompts. For output-based unlearning methods, it introduces a diversified prompting strategy that replaces the target keyword with contextualized target prompts and contextualized anchor prompts. For attention-based methods, it proposes a diversified embedding mixup strategy that mixups target keyword embeddings and their contextualized counterparts.
Experiments across five concept-erasure scenarios (i.e., celebrities, objects, copyrighted characters, nudity, and artistic styles) show that integrating the proposed method with five existing unlearning frameworks consistently improves erasure performance.

**Strengths:**

- The proposed method is simple yet effective in improving concept erasure performance.
- The paper conducts comprehensive experiments across five unlearning domains and provides a detailed analysis of prompt diversity, exploring contextual complexity from level 0 to level 7 and varying the number of prompts from 5 to 50.
- The paper is well-written and clear. The example in lines 239-242 helps me understand the diversified embedding mixup mechanism.

**Weaknesses:**

- The method shows limited effectiveness in concept preservation. In the copyrighted concept erasure task (Table 1 right), preservation improvements are marginal, about 0.02 in LPIPS for ESD, AP, AGE, and ACE, and less than 1 point in CLIP-i, CLIP-t, and GPT scores. Similarly, in the nudity erasure task (Table 2), preservation improvements in FID and CLIP metrics are negligible or even degrade.
- While the paper mentions adversarially trained unlearning approaches such as AdvUnlearn, R.A.C.E., Receler, or RECE, it does not compare against them experimentally. I’m interested in whether learning from soft prompts (as in those works) or learning from pre-defined textual prompts (as in this paper) leads to better robustness, so including those works in the experiments could provide better understanding.
- Regarding the recovery attack experiments (sec. 4.4), details like the contextual level of prompts used and the total number of prompts are missing.

**Questions:**

- How sensitive is the mixup hyper-parameter $\alpha$ in the diversified embedding mixup? For example, will the method remain effective for attention-based unlearning if $\alpha$ is reduced (e.g., to 0.6)?
- Any insights or ideas on why the erasure performance drops a lot when level-4 prompts are used (Fig. 4a)?
- In the ablation on the number of prompts (Fig. 4b), would the optimal number of prompts decrease if higher-complexity prompts (e.g., level-3) were used?

---

> ### Author Response · Authors · 2025-11-22
> **Author's response**
>
> We thank the reviewer for acknowledging our strengths and providing constructive comments. We would like to address the remaining concerns as follows:
>
> **Q1. How sensitive is the mixup hyper-parameter $\alpha$ in the diversified embedding mixup? For example, will the method remain effective for attention-based unlearning if $\alpha$ is reduced (e.g., to 0.6)?**
>
> We follow the reviewer’s suggestion and analyze the impact of the mixup hyper-parameter $\alpha$ on the performance of diversified embedding mixup. The experiment is conducted on the 10-celebrity erasure task, using the settings described in Section 4.1 and Appendix B.1.1. As shown in the table, there is a clear trade-off between unlearning performance and preservation performance.
>
> More specifically, reducing $\alpha$ increases the contribution of the diversified contextual concepts in the mixed embedding. This leads to stronger entanglement between the target concept and the surrounding context, causing preservation performance to drop significantly. Conversely, unlearning performance initially increases, however, quickly saturates around $\alpha = 0.95$, indicating that overly diverse mixed embeddings eventually make it difficult for the model to correctly identify the target concept, thus limiting further gains.
>
> In our main experiments, we set $\alpha = 0.999$ to maintain a strong balance between unlearning effectiveness and preservation quality.
>
> | **Method**| **Alpha weight** | **Erasure** || **Preservation** ||||
> |-|-|-|-|-|-|-|-|
> ||| CLIP-i↓| GPT-score↓ | LPIPS↓| CLIP-i↑ | CLIP-t↑ | GPT-score↑|
> | UCE| -| 60.59      | 4.48| 0.65| 62.90| 23.97| **31.42**|
> | **Diversified-UCE Mixup**| 0.999| 58.55| 2.83| **0.64**| **66.92** | **24.57** | 24.40|
> | Diversified-UCE Mixup| 0.99| 58.01| 2.79| **0.64**| 64.60| 23.95| 23.00|
> | Diversified-UCE Mixup| 0.95| **57.25**| 3.05| 0.65| 60.58| 22.32| 15.37|
> | Diversified-UCE Mixup| 0.90| 57.66| 2.53| 0.66| 60.09| 21.88| 13.10|
> | Diversified-UCE Mixup| 0.80| 57.45| **2.46**   | 0.67| 57.84| 20.86| 10.07|
>
>
> **Q2. Any insights or ideas on why the erasure performance drops a lot when level-4 prompts are used (Fig. 4a)?**
>
> We observe a significant drop in erasure performance when using Level-4 fine-tuning prompts, whose complexity is substantially higher than Levels 1-3. This degradation can be explained from two complementary perspectives:
> -  **Output-based behavior under highly complex prompts**. Level-4 prompts contain a large number of contextual attributes. For output-based unlearning methods such as ESD, the generated image must reflect many elements specified in the prompt. As these elements occupy substantial visual capacity, the presence of the target concept becomes inherently weaker. Consequently, the model only needs to perform a minor removal for the concept to *'vanish'*, or in some cases, the target concept is already faint in the output before unlearning. In contrast, simpler prompts in Levels 1-3 encourage the target concept to appear more clearly, yielding more stable and interpretable erasure outcomes.
> - **Signal dilution during the denoising process**. During denoising, simple prompts such as *'A photo of Henry Cavill waving'* (Level 1) or *'A photo of Henry Cavill waving in a crowded room'* (Level 2) reinforce the identity signal at nearly every timestep. However, for a highly complex prompt such as *'A photo of Henry Cavill jogging across a bridge at dawn beside a man wearing a campaign T-shirt'*, the model must allocate denoising capacity to render competing contextual cues, including the bridge, the dawn lighting, the jogging motion, the additional person, and clothing details. At various timesteps, the identity signal associated with *'Henry Cavill'* becomes overshadowed by these competing contexts. As the target-concept signal is diluted throughout the denoising trajectory, the effective erasure signal weakens, leading to reduced unlearning performance.

---

> ### Author Response · Authors · 2025-11-22
> **Author's response**
>
> **Q3. In the ablation on the number of prompts (Fig. 4b), would the optimal number of prompts decrease if higher-complexity prompts (e.g., level-3) were used?**
>
> In the experiment illustrated in Fig.4a, we fine-tune four models using four prompt sets with increasing complexity levels (Level 1-4), each containing 20 prompts. From the complementary analysis in Fig.4b, we observe that for Level-1 prompts, the number of prompts has only a *minor effect* on overall performance: the proposed method consistently outperforms the baseline in both erasure and preservation, with 10 prompts serving as an empirically optimal choice.
>
> This observation suggests that when prompt complexity increases to Levels 2 or 3, the number of prompts is also likely to have only a limited impact on the final performance. Fig.4a shows that a set of 20 Level 2 or Level 3 prompts already achieves outperform results compared to the baseline. This implies that further reducing the number of prompts -- similar to the setup in Fig.4b - should maintain similar performance. Therefore, the primary source of performance gains originates not from the sheer quantity of prompts, but from the enhanced prompt diversity and the controlled complexity of contextual variations introduced during fine-tuning.

---

> ### Author Response · Authors · 2025-11-22
> **Author's responses**
>
> **Weakness 1. Limited effectiveness in concept preservation in copyrighted and nudity erasure tasks.**
>
>
> Our method is primarily designed to improve the *precision of erasure* by disentangling the target concept from its unintended associated concepts. This more accurate concept identification enhances both erasure effectiveness and robustness. We validate these aspects extensively across five erasure tasks—celebrities, objects, copyrighted characters, artistic styles, and nudity (Table 4)—and further evaluate robustness on three tasks (Table 3). Our approach achieves substantial gains, including improvements of 16\% against Ring-A-Bell, 5\% against Indirect Recovery (ICLR 2025), and 25\% against Noise-Based Attack (recently accepted to NeurIPS 2025).
>
> Although we do not modify the preservation loss, a key strength of our framework is that **contextual disentanglement naturally improves preservation**. By removing only the target concept and retaining its accompanying context, we reduce unintended interference with unrelated concepts. This effect leads to consistent preservation improvements on celebrities (Table 1), artistic styles (Table 10), and even modest gains on copyrighted characters.
>
> Regarding the nudity erasure task specifically, we acknowledge that preservation performance shows a slight drop. Importantly, this outcome is *expected and inherent to the task*. Nudity prompts in I2P and Ring-A-Bell are explicitly designed to elicit highly entangled visual elements (e.g., skin exposure, pose, body structure), many of which overlap with the contextual cues needed for preservation. As a result, removing nudity often requires suppressing visual patterns that are difficult to disentangle from the surrounding context, making perfect preservation fundamentally more challenging for all methods, including ours. Even so, our approach still improves *erasure accuracy and robustness* more than existing baselines, which is the primary aim of nudity unlearning.
>
> In summary, while nudity erasure inherently involves stronger concept entanglement that limits preservation, our method still demonstrates clear preservation gains across most tasks and offers substantial improvements in targeted removal and robustness—the central goals of concept unlearning.
>
> **Weakness 2. Additional experiments with adversarially trained unlearning approaches.**
>
> We follow the reviewer's suggestion to conduct an additional experiment with Receler (erasing the celebrity *'Henry Cavill'*) and provide the results in the table below.
>
> | **Method**| **Erasure** || **Preservation** ||||
> |-|-|-|-|-|-|-|
> || CLIP-i↓    | GPT-score↓ | LPIPS↓| CLIP-i↑ | CLIP-t↑ | GPT-score↑       |
> | Receler| **51.61**| **0.00**| 0.57| 73.93   | 25.27  | 47.07           |
> | **Diversified-Receler**| 52.90| **0.00**| **0.52**| **77.65** | **26.17** | **55.60** (+8.53%) |
>
> As shown in the table above, integrating Diversified Unlearning into Receler preserves the original model’s erasure performance while markedly enhancing preservation, yielding an 8.53\% improvement in the GPT-score. Beyond the quantitative metrics, we also provide qualitative examples in the anonymous GitHub repository to enable direct visual comparison between Receler and Diversified-Receler. Although additional fine-tuning of our add-on module could potentially lead to even stronger performance, we intentionally employ the same default configuration used in our other experiments to ensure fairness and consistency.
>
> These additional results—together with the comprehensive experiments conducted on recent state-of-the-art unlearning methods such as *ACE* (CVPR 2025) and *AGE* (ICLR 2025)—further reinforce both the effectiveness and the broad applicability of our proposed approach.

---

> ### Author Response · Authors · 2025-11-22
> **Author's responses**
>
> **Weakness 3. Regarding the recovery attack experiments (sec. 4.4), details like the contextual level of prompts used and the total number of prompts are missing.**
>
> Thanks the review for pointing it out. Below are the detail settings for recovery attacks, we will update them into the revised paper.
>
> - **Noise-based attack**: We compared the performance of AP and Diversified-AP using an evaluation set of celebrity identities. Specifically, the set contains 400 prompts, covering 10 celebrities with 8 levels per individual. All of the designed prompt levels are presented in Appendix B.1.1.
>
> - **Indirect recovery**: We use the prompt: *"Short stocky Italian plumber, red hat, thick rounded mustache, blue overalls, white gloves in Mushroom Kingdom cartoon with Nintendo style"*, together with 200 seeds containing related keywords, which can cause the model to recall the target without directly mentioning its name as proposed in the paper [1]. These prompts are used to evaluate the ability to remove copyrighted characters, and the results are visually presented in Table 9.
>
> - **Adversarial prompts**: The prompts from Ring-A-Bell were selected for evaluating the performance of models designed for explicit-content unlearning.
>
> **References**: [1] Luxi He, Yangsibo Huang, Weijia Shi, Tinghao Xie, Haotian Liu, Yue Wang, Luke Zettlemoyer, Chiyuan Zhang, Danqi Chen, and Peter Henderson. Fantastic copyrighted beasts and how (not) to generate them. arXiv preprint arXiv:2406.14526, 2024

---

> > ### Author Response · Authors · 2025-11-27
> >
> > Dear Reviewer **WWWB**,
> >
> > We would like to provide additional experiments on the suggested adversarial-training–based unlearning methods, including **AdvUnlearn**, **RECE**, and **Receler** (previously provided, but reproduced here for clarity). AdvUnlearn and Receler belong to the *output-based unlearning* family, following the ESD framework, while RECE is an *attention-based* method with a closed-form solution similar to UCE.
> >
> > We integrate our proposed diversification technique into these keyword-based frameworks, resulting in three diversified variants: **Diversified-AdvUnlearn**, **Diversified-RECE**, and **Diversified-Receler**. We evaluate them on two erasure tasks: removing the celebrity *Henry Cavill* and removing *ImageNette* objects, using the same experimental settings detailed in Appendix B.1.1. The code has also been updated in the anonymous repository for further inspection.
> >
> > From the first table, we observe that:
> >
> > 1. **Compared to Receler**, our diversified counterpart achieves *equally strong unlearning performance* (both methods reach zero GPT-score and very low CLIP-i), while achieving *substantially better preservation performance*, improving by **+8.53 GPT-score** or **+3.7 CLIP-i**.
> > 2. **Compared to RECE**, our diversified variant achieves slightly better preservation performance, along with a noticeably improved unlearning performance.
> >
> > The second table shows that, relative to AdvUnlearn, our Diversified-AdvUnlearn variant achieves *comparable unlearning effectiveness* while providing a *much stronger preservation ability*, including a notable **+30% PSR-5 improvement** on the task of erasing five objects simultaneously.
> >
> > Taken together, these results—along with the extensive experiments on recent state-of-the-art methods such as **ACE (CVPR 2025)** and **AGE (ICLR 2025)**—further strengthen our claim regarding both the *effectiveness* and the *generality* of our diversified unlearning approach. Our method not only enhances unlearning quality but also consistently improves preservation across diverse keyword-based unlearning frameworks.
> >
> > | **Method**| **Erasure** || **Preservation** ||||
> > |-|-|-|-|-|-|-|
> > || CLIP-i↓    | GPT-score↓ | LPIPS↓| CLIP-i↑ | CLIP-t↑ | GPT-score↑|
> > | **Receler**| **51.61**| **0.00**| 0.57| 73.93   | 25.27  | 47.07|
> > | **Diversified-Receler**| 52.90| **0.00**| **0.52**| **77.65** | **26.17** | **55.60** (+8.53%) |
> > | **RECE**| 59.42| 8.05| **0.79**| 77.34| 29.32| 77.42|
> > | **Diversified-RECE**| **58.57**| **7.78**| **0.79**| **77.95**| **29.64**| **77.87**|
> >
> > Quantitative results of erasing `Henry Cavill' using baseline Receler and RECE compared with the diversified counterpart, as detailed in Appendix B.1.1.
> >
> > | **Object**| **'Garbage Truck' erasure**|||| **'Five objects' erasure**||||
> > |-|-|-|-|-|-|-|-|-|
> > || ESR-1| ESR-5|PSR-1|PSR-5|ESR-1|ESR-5|PSR-1|PSR-5|
> > | **AdvUnlearn**|**1.000**|**0.992**|0.768|0.936|**1.000**|**0.990**|0.459|0.604|
> > | **Diversified-AdvUnlearn**|0.996|0.976|**0.807**|**0.952**|0.988|0.974|**0.732**| **0.916** |
> >
> > Quantitative results of object erasure using keyword-based methods compared with their diversified counterparts, evaluated across prompts of varying complexity levels. Erasure Success Rate (ESR) and Preservation Success Rate (PSR), computed with a pre-trained ResNet-50, where higher values indicate better performance. Overall, under the setting of simultaneously erasing five objects.

---

> > > ### Author Response · Authors · 2025-11-27
> > >
> > > Dear Reviewer **WWWB**,
> > >
> > > Thank you once again for the time and care you invested in reviewing our submission. We sincerely appreciate your constructive feedback, which has helped us further strengthen the paper. We are writing to kindly remind you that we have now posted a detailed response to your comments.
> > >
> > > In our rebuttal, we have specifically addressed the key concerns you raised:
> > >
> > > #### **1. The impact of the hyperparameter α in Diversified Embedding Mixup**
> > > Our experiments examining different values of α show that increasing the contextual influence injected into the target concept leads to a noticeable decline in preservation performance, while yielding only marginal gains in erasure effectiveness. We therefore adopt α = 0.999 throughout the paper, as it provides a well-balanced trade-off between unlearning strength and preservation quality.
> > >
> > > #### **2. The effect of prompt complexity and prompt quantity**
> > > In Section 4.5 and Appendix C.6, we present an ablation study analyzing how contextual diversity and semantic complexity influence the performance of our method, leading to three key observations:
> > > (i) diversity is essential,
> > > (ii) moderate contextual complexity is most effective, and
> > > (iii) increasing the number of prompts yields diminishing returns.
> > > When using highly complex Level-4 prompts for fine-tuning, the influence of the target concept in the generated images becomes substantially weaker due to the dominance of surrounding contextual semantics. This reduced prominence of the target concept directly contributes to the observed degradation in erasure effectiveness.
> > >
> > > #### **3. Enhancing adversarial training–based unlearning methods**
> > > Our method functions as an **add-on module** that can be seamlessly integrated into existing unlearning baselines. When applied to adversarial training–based approaches such as **Receler**, our method enhances preservation while maintaining the strong erasure performance intrinsic to these adversarial baselines. Similar improvements are also observed in our new experiments with **AdvUnlearn** and **RECE**.
> > >
> > > ---
> > >
> > > We would be sincerely grateful if you could take a moment to review our detailed responses. Please feel free to reach out with any additional questions during the remaining days of the discussion period — we are happy to provide further clarification if needed.
> > >
> > > Thank you again for your valuable insights.
> > >
> > > **Best regards,**
> > > *The Authors of Submission 3912*

---

> ### Comment · Reviewer_WWWB · 2025-11-28
> **Official Comment by Reviewer WWWB**
>
> Thank you for your response to my questions and concerns. I still have one question.
>
> Regarding the second weakness, what I was hoping to see is a comparison between the gains achieved by adversarially learned soft prompts (e.g., in Receler) and the gains achieved by your diversified hard prompts. I was not suggesting adding your method on top of those approaches. Take Receler for example, if Receler’s adversarial prompt learning method were replaced with your diversified prompts or diversified embedding mixup, how would its erasure and preservation performance change?
> Since both approaches aim to fully erase a target concept but differ in whether they learn soft vs. hard prompts. It would be helpful to understand which one provides better erasure strength and robustness.
>
> One minor thing is that from your additional experiment in the response to reviewer fJw7, Noise-based UCE mixup significantly improves preservation. I’m wondering whether carefully learned soft prompts could similarly achieve robust erasure with strong preservation in unlearned diffusion models.

---

> > ### Author Response · Authors · 2025-12-02
> > **Official Comment by Authors**
> >
> > We thank the reviewer for carefully reading our rebuttal, and we are glad that most of the reviewer's concerns have been addressed. For the single remaining concern, we would like to respond as follows:
> >
> > **1. Regarding the second weakness, what I was hoping to see is a comparison between the gains achieved by adversarially learned soft prompts (e.g., in Receler) and the gains achieved by your diversified hard prompts. I was not suggesting adding your method on top of those approaches. Take Receler for example, if Receler’s adversarial prompt learning method were replaced with your diversified prompts or diversified embedding mixup, how would its erasure and preservation performance change? Since both approaches aim to fully erase a target concept but differ in whether they learn soft vs. hard prompts. It would be helpful to understand which one provides better erasure strength and robustness.**
> >
> > Firstly, we sincerely apologize for misunderstanding your question. Since our method is designed as an *add-on module* that enhances existing baseline unlearning methods rather than serving as an independent, fully self-contained framework, we initially assumed that the question was asking for a comparison between Receler and our diversified counterpart.
> > Regarding that perspective, where our diversified method can be integrated directly to adversarial training keyword-based approach, it has been shown clearly in this rebuttal that our method can significantly improve the keyword based counterparts in preservation performance while achieve competitive unlearning performance. We would like to restate the results for better reading comprehension.
> >
> > Firstly, we sincerely apologize for misunderstanding your question. Since our method is designed as an *add-on module* that enhances existing baseline unlearning methods rather than serving as an independent, fully self-contained framework, we initially assumed that the question was asking for a comparison between Receler and our diversified counterpart. Under this perspective---our diversified method can be integrated directly to adversarial training keyword-based approach---the results presented in the rebuttal already demonstrate that our method can substantially boost preservation performance while maintaining competitive unlearning strength. For clarity, we restate these results below for easier reference.
> >
> > ### *Table: 'Henry Cavill' Concept Under RECE and Receler compared with their Diversified methods.*
> >
> > | **Method**| **Erasure** || **Preservation** ||||
> > |-|-|-|-|-|-|-|
> > || CLIP-i↓    | GPT-score↓ | LPIPS↓| CLIP-i↑ | CLIP-t↑ | GPT-score↑|
> > | **Receler**| **51.61**| **0.00**| 0.57| 73.93   | 25.27  | 47.07|
> > | **Diversified-Receler**| 52.90| **0.00**| **0.52**| **77.65** | **26.17** | **55.60** (+8.53%) |
> > | **RECE**| 59.42| 8.05| **0.79**| 77.34| 29.32| 77.42|
> > | **Diversified-RECE**| **58.57**| **7.78**| **0.79**| **77.95**| **29.64**| **77.87**|
> >
> > ### *Table: Imagenette Concept Under AdvUnlearn and its Diversified variant.*
> >
> > | **Object**| **'Garbage Truck' erasure**|||| **'Five objects' erasure**||||
> > |-|-|-|-|-|-|-|-|-|
> > || ESR-1| ESR-5|PSR-1|PSR-5|ESR-1|ESR-5|PSR-1|PSR-5|
> > | **AdvUnlearn**|**1.000**|**0.992**|0.768|0.936|**1.000**|**0.990**|0.459|0.604|
> > | **Diversified-AdvUnlearn**|0.996|0.976|**0.807**|**0.952**|0.988|0.974|**0.732**| **0.916** |
> >
> > Secondly, intuitively, we believe there is a fundamental difference between diversified token mixup and adversarially learned soft prompts that distinguishes our approach from prior methods.
> >
> > In our diversified token mixup (contextualized or noise-based), only the tokens associated with the target concept (e.g., "\<Henry\> \<Cavill\>'') are diversified by injecting additional noise $n$ (or contextualized prompts $c$), as described in the equation below:
> >
> > $$ f(τ(c_e), n)^i = α τ(c_e)^i + (1 − α) n \text{ if token } i \text{ belongs to } c_e, \text{ and } f(τ(c_e), n)^i = τ(c_e)^i  \text{ otherwise}. $$
> >
> > This design ensures that all non-target tokens remain unchanged.
> >
> > In contrast, adversarially learned soft prompts---as used in RECE or Receler---update *all* tokens, regardless of their semantic role. As a result, these methods often inject unnecessary perturbations into the entire prompt embedding, which can unintentionally distort the semantic structure and hinder preservation.
> >
> > Compared to such approaches, our diversified token mixup, despite its simplicity, strategically limits perturbation to only the target tokens. This keeps the surrounding contextual tokens intact, avoids introducing excessive noise, and preserves sufficient information for the encoder to reliably identify the target concept while still introducing the diversity needed to facilitate effective unlearning.
> >
> > We believe that our simple yet effective diversified token mixup provides a more controlled and semantically focused mechanism for improving the trade-off between erasure and preservation, ultimately leading to a more stable and robust unlearning process.

---

> ### Author Response · Authors · 2025-12-02
> **Official Response by Authors**
>
> Thirdly, when revisiting the results of Diversified UCE---which can be viewed as an analogue of “diversified RECE with hard prompts”---we clearly observe that diversified hard prompts deliver markedly stronger preservation than adversarially learned soft prompts. Notably, we observe improvements of 13 percentage points in GPT-score, 9 points in CLIP-i score, and 0.3 in LPIPS. These consistent and substantial gains demonstrate that diversified hard prompts provide a far more stable and reliable preservation behavior, without sacrificing competitive erasure performance.
>
> In conclusion, considering (i) the clarified experimental evidence, (ii) the conceptual distinction between localized (target-only) perturbations and global adversarial updates, and (iii) the strong empirical improvements achieved by diversified hard prompts, we believe that diversified prompts offer a more robust, effective, and semantically controlled alternative to adversarially learned soft prompts for achieving reliable erasure while preserving non-target content.
>
> **2. One minor thing is that from your additional experiment in the response to reviewer fJw7, Noise-based UCE mixup significantly improves preservation. I’m wondering whether carefully learned soft prompts could similarly achieve robust erasure with strong preservation in unlearned diffusion models.**
>
> We thank the reviewer for the interesting hypothesis. We would like to restate the answer above for better reading comprehension. While we have not conducted explicit experiments on learned soft prompts, we intuitively believe there is a fundamental difference between noise-based UCE mixup and adversarially learned soft prompts that distinguishes our approach from prior methods.
>
> In noise-based UCE mixup, only the tokens associated with the target concept (e.g., "\<Henry\> \<Cavill\>'') are diversified by injecting additional noise $n$, as described in the equation below:
>
> $$ f(τ(c_e), n)^i = α τ(c_e)^i + (1 − α) n \text{ if token } i \text{ belongs to } c_e, \text{ and } f(τ(c_e), n)^i = τ(c_e)^i  \text{ otherwise}. $$
>
> This design ensures that all non-target tokens remain unchanged.
>
> In contrast, adversarially learned soft prompts---as used in RECE or Receler---update *all* tokens, regardless of their semantic role. As a result, these methods often inject unnecessary perturbations into the entire prompt embedding, which can unintentionally distort the semantic structure and hinder preservation.
>
> | **Method**                         | **CLIP-i↓** | **GPT-score↓** | **LPIPS↓** | **CLIP-i↑** | **CLIP-t↑** | **GPT-score↑** |
> |-----------------------------------|-------------|----------------|------------|-------------|-------------|----------------|
> | RECE (learned soft prompt)        | **59.42**   | 8.05           | 0.79       | 77.34       | 29.32       | 77.42          |
> | RECE (diversified hard prompt)    | 61.23       | **6.95**       | **0.48**   | **86.11**   | **29.95**   | **90.95**      |
>
> Compared to such approaches, our diversified token mixup, despite its simplicity, strategically limits perturbation to only the target tokens. This keeps the surrounding contextual tokens intact, avoids introducing excessive noise, and preserves sufficient information for the encoder to reliably identify the target concept while still introducing the diversity needed to facilitate effective unlearning.
>
> We believe that our simple yet effective diversified token mixup provides a more controlled and semantically focused mechanism for improving the trade-off between erasure and preservation, ultimately leading to a more stable and robust unlearning process.

---

### Official Review · Reviewer_sj5a · 2025-10-31

**Soundness:** 4
**Presentation:** 3
**Contribution:** 3
**Rating:** 6
**Confidence:** 3

**Summary:**

This paper addresses the limitations of current unlearning approaches that focus only on removing target concepts associated with specific keywords. Such keyword-based unlearning often struggles to effectively erase the full concept and tends to harm unrelated concepts. To overcome these limitations, the authors propose Diversified Unlearning, a distributional unlearning method that leverages LLM-generated diversified prompting. By replacing a single target keyword with a set of contextualized prompts, this method broadens the erasure scope, strengthens resistance against recovery attacks, and mitigates damage to unrelated concepts.

**Strengths:**

1. The paper shifts the perspective from keyword-based unlearning to distributional unlearning, offering a novel conceptualization of representing concepts as distributions. This perspective provides a promising direction for developing more robust unlearning techniques.
2. The proposed approach functions as a plug-in module compatible with existing unlearning methods, effectively enhancing their performance.
3. Extensive experiments on Stable Diffusion 1.4 demonstrate the effectiveness of the proposed method, achieving superior results compared with prior approaches.

**Weaknesses:**

1. Although distributional unlearning is an insightful perspective, the paper lacks a systematic exploration of how distributional ranges differ across concept categories (e.g., celebrity, copyrighted character, explicit concept), and how corresponding contextualized prompts should be designed for each category.
2. The improvement of the proposed method over robust unlearning methods such as [1][2] is not sufficiently demonstrated.
3. All experiments are conducted on Stable Diffusion 1.4, leaving uncertainty about generalization to more recent models (e.g., Stable Diffusion 3.5, FLUX).

Minor：
1. The description of the preprocessing and postprocessing steps (lines 56–57) seems disconnected from the keyword-based unlearning process.

[1] Zhang Y, Chen X, Jia J, et al. Defensive unlearning with adversarial training for robust concept erasure in diffusion models[J]. Advances in neural information processing systems, 2024, 37: 36748-36776.
[2] Gong C, Chen K, Wei Z, et al. Reliable and efficient concept erasure of text-to-image diffusion models[C]//European Conference on Computer Vision. Cham: Springer Nature Switzerland, 2024: 73-88

**Questions:**

1. Could the authors provide an ablation study on the hyperparameter $\alpha$ introduced in Diversified Embedding Mixup to clarify its impact on performance?
2. Why did the authors choose GPT-Score over identity similarity as the evaluation metric in Section 4.1?
3. Table 10 shows that unlike other concepts (e.g., identity), the proposed method fails to improve artistic style unlearning. Could the authors clarify why this occurs?

---

> ### Author Response · Authors · 2025-11-22
>
> We thank the reviewer for the positive feedback on the novelty of our method as well as the practical and effectiveness of our method demonstrated on comprehensive set of experiments. We would like to address the remaining concerns as follows:
>
> **W1. Although distributional unlearning is an insightful perspective, the paper lacks a systematic exploration of how distributional ranges differ across concept categories (e.g., celebrity, copyrighted character, explicit concept), and how corresponding contextualized prompts should be designed for each category.**
>
> In Section 4.5 and Appendix C.6, we provide an ablation study analyzing how the *diversity* and *contextual complexity* of contextualized prompts affect the performance of our method. Although this study is not exhaustively repeated for every concept category, the findings reveal clear and generalizable principles that guide effective prompt design across different distributional ranges. The key observations can be summarized as follows:
>
> - **Diversity is essential.** Even simple diversified prompts consistently improve both erasure and preservation relative to the baseline, demonstrating that incorporating contextual variation is crucial regardless of the concept category.
>
> - **Moderate contextual complexity is most effective.** While added context strengthens the disentanglement between the target concept and its associated attributes, overly complex contexts introduce noise and degrade erasure performance (see Fig. 4(a)).
>
> - **Prompt quantity exhibits diminishing returns.** Under moderate contextual complexity, increasing the number of distinct prompts from 5 to 50 reliably outperforms baselines in both erasure and preservation. However, approximately **10 prompts** achieves the best trade-off between gain and efficiency (see Fig. 4(b)).
>
> Although we did not replicate this ablation for every unlearning scenario (e.g., nudity, copyrighted characters), these observations align closely with the distributional characteristics of the categories studied and generalize well in practice. Most importantly, these principles directly contribute to the consistent performance gains we observe across all concept categories in our experiments.
>
> **W2. Additional experiments on robust unlearning methods.**
>
> **Answer:**
> We follow the reviewer's suggestion to conduct an additional experiment with Receler, an adversarial training unlearning approach under experiment of erasing 'Henry Cavill' as detailed in Appendix B.1.1 and provide the results in the table below.
>
> | **Method**| **Erasure** || **Preservation** ||||
> |-|-|-|-|-|-|-|
> || CLIP-i↓    | GPT-score↓ | LPIPS↓| CLIP-i↑ | CLIP-t↑ | GPT-score↑       |
> | Receler| **51.61**| **0.00**| 0.57| 73.93   | 25.27  | 47.07           |
> | **Diversified-Receler**| 52.90| **0.00**| **0.52**| **77.65** | **26.17** | **55.60** (+8.53%) |
>
> As shown in the table above, integrating Diversified Unlearning into Receler preserves the original model’s erasure performance while markedly enhancing preservation, yielding an 8.53% improvement in the GPT-score. Beyond the quantitative metrics, we also provide qualitative examples in the anonymous GitHub repository to enable direct visual comparison between Receler and Diversified-Receler. Although additional fine-tuning of our add-on module could potentially lead to even stronger performance, we intentionally employ the same default configuration used in our other experiments to ensure fairness and consistency.
>
> These additional results—together with the comprehensive experiments conducted on recent state-of-the-art unlearning methods such as *ACE* (CVPR 2025) and *AGE* (ICLR 2025)—further reinforce both the effectiveness and the broad applicability of our proposed approach.
>
> **W3. All experiments are conducted on Stable Diffusion 1.4, leaving uncertainty about generalization to more recent models (e.g., Stable Diffusion 3.5, FLUX).**
>
> Our experimental setup follows the standard evaluation protocol widely adopted in the machine unlearning literature. Notably, the majority of recent state-of-the-art unlearning methods—including *ACE* (CVPR 2025), *AGE* (ICLR 2025), as well as the comprehensive investigation in [1] (NeurIPS 2025)—are all evaluated on Stable Diffusion 1.4 for fair comparison and methodological consistency. Using the same backbone ensures that improvements introduced by our method are attributable to the unlearning mechanism itself rather than architectural differences across model generations.
>
> While our current work focuses on SD 1.4 to remain comparable with prior art, our proposed approach is model-agnostic and modular—requiring only access to the text encoder (diversified embedding mixed variant). Thus, extending the method to more recent diffusion models (e.g., SD 3.5, FLUX) is straightforward, and we view this as promising future work for broader benchmarking.
>
> **References:**
> [1] Lu, Kevin, et al. "When Are Concepts Erased From Diffusion Models?." NeurIPS 2025.

---

> ### Author Response · Authors · 2025-11-22
>
> **W4. The description of the preprocessing and postprocessing steps (lines 56–57) seems disconnected from the keyword-based unlearning process.**
>
> We thank the reviewer for pointing this out. This part is intended to provide a brief motivation for why concept unlearning offers a more principled solution compared to preprocessing or postprocessing heuristics. We will revise the text to strengthen the connection and improve clarity in the updated version.
>
> **Q1. Could the authors provide an ablation study on the hyperparameter introduced in Diversified Embedding Mixup to clarify its impact on performance?**
>
> We follow the reviewer’s suggestion and analyze the impact of the mixup hyper-parameter $\alpha$ on the performance of diversified embedding mixup. The experiment is conducted on the 10-celebrity erasure task, using the settings described in Section 4.1 and Appendix B.1.1. As shown in the table, there is a clear trade-off between unlearning performance and preservation performance.
>
> More specifically, reducing $\alpha$ increases the contribution of the diversified contextual concepts in the mixed embedding. This leads to stronger entanglement between the target concept and the surrounding context, causing preservation performance to drop significantly. Conversely, unlearning performance initially increases, however, quickly saturates around $\alpha = 0.95$, indicating that overly diverse mixed embeddings eventually make it difficult for the model to correctly identify the target concept, thus limiting further gains.
>
> In our main experiments, we set $\alpha = 0.999$ to maintain a strong balance between unlearning effectiveness and preservation quality.
>
> | **Method**| **Alpha weight** | **Erasure** || **Preservation** ||||
> |-|-|-|-|-|-|-|-|
> ||| CLIP-i↓| GPT-score↓ | LPIPS↓| CLIP-i↑ | CLIP-t↑ | GPT-score↑|
> | UCE| -| 60.59      | 4.48| 0.65| 62.90| 23.97| **31.42**|
> | **Diversified-UCE Mixup**| 0.999| 58.55| 2.83| **0.64**| **66.92** | **24.57** | 24.40|
> | Diversified-UCE Mixup| 0.99| 58.01| 2.79| **0.64**| 64.60| 23.95| 23.00|
> | Diversified-UCE Mixup| 0.95| **57.25**| 3.05| 0.65| 60.58| 22.32| 15.37|
> | Diversified-UCE Mixup| 0.90| 57.66| 2.53| 0.66| 60.09| 21.88| 13.10|
> | Diversified-UCE Mixup| 0.80| 57.45| **2.46**   | 0.67| 57.84| 20.86| 10.07|

---

> > ### Author Response · Authors · 2025-11-22
> >
> > **Q2. Why did the authors choose GPT-Score over identity similarity as the evaluation metric in Section 4.1?**
> >
> > **Answer:**
> > For evaluating unlearning on celebrity concepts, we employ metrics such as LPIPS and CLIP, complemented by the *GPT-score*. The decision to incorporate GPT-score is motivated by several factors:
> >
> > - It has been demonstrated to be effective in the context of celebrity images, as evidenced by Dreambench++ [1], which provides a human-aligned benchmark for personalized image generation.
> > - GPT-score is a generalizable metric that can also be applied to copyrighted concepts where non-facial attributes, such as hairstyle or partial clothing, play an important role in defining the target concept.
> > - While conventional face recognition models could be considered, these models are typically trained to classify highly similar faces (e.g., siblings or relatives) [2]. However, generative diffusion models often produce variations in facial details that would be perceived as different identities by a face recognition model, even though the output still reflects the intended target concept when considering additional attributes. GPT-score, by comparing a generated image to a reference sample in a flexible manner, is therefore more suitable for evaluating whether an unlearning method effectively removes the target concept while preserving contextual fidelity.
> >
> > - Moreover, most public face recognition models are trained on large-scale datasets of real images to achieve strong performance [2]. Although some recent works explore synthetic data [3, 4], these images are typically produced by face-specialized generative models rather than general-purpose diffusion models. Applying such public models to evaluate unlearning in general-purpose diffusion models would require fine-tuning these face recognition models to align with the new input domain. This additional adaptation step significantly limits their practicality and reduces their applicability, especially when compared to GPT-score.
> >
> > **References:**
> > [1] Y. Peng, Y. Cui, H. Tang, Z. Qi, R. Dong, J. Bai, C. Han, Z. Ge, X. Zhang, and S.-T. Xia. *Dreambench++: A human-aligned benchmark for personalized image generation.* In *ICLR*, 2025. URL: [https://dreambenchplus.github.io/](https://dreambenchplus.github.io/)
> >
> > [2] Guo J., Deng J., An X., Yu J., Gecer B. (2025). InsightFace: State-of-the-art 2D and 3D Face Analysis Project. GitHub repository: [https://github.com/deepinsight/insightface](https://github.com/deepinsight/insightface)
> >
> > [3] Wu, H., Singh, J., Tian, S., Zheng, L., and Bowyer, K. W. (2025). Vec2Face: Scaling face dataset generation with loosely constrained vectors. International Conference on Learning Representations (ICLR).
> >
> > [4] Lin, X., Huang, Y., Xu, J., Mi, Y., Zhou, S., & Ding, S. (2025). “UIFace: Unleashing Inherent Model Capabilities to Enhance Intra-Class Diversity in Synthetic Face Recognition.” International Conference on Learning Representations (ICLR).
> >
> > **Q3. Table 10 shows that unlike other concepts (e.g., identity), the proposed method fails to improve artistic style unlearning. Could the authors clarify why this occurs?**
> >
> > **Answer:**
> >
> > Overall, our method demonstrates positive results in preservation but exhibits lower erasure performance on the `Van Gogh` concept across various unlearning methods. Nevertheless, *Diversified Unlearning* still achieves encouraging outcomes on the `Kelly McKernan' concept under certain settings. We hypothesize several factors to explain these observations:
> >
> > - **Abstract nature of artistic style.** Artistic style concepts are inherently abstract, and accurate evaluation may sometimes require human judgment. For instance, while CLIP-t measures the similarity between generated images and input prompts, it may produce counterintuitive results:
> >   1. An image that preserves content but loses the targeted style could score higher compared to
> >   2. An image whose content deviates but the style is not fully removed.
> >   This indicates that (1) has lower erasure performance than (2).
> >
> > - **Challenges in constructing diversified prompts for artistic style.** For output-based methods such as ESD and AP, we use GPT to construct prompts like *`A work of art of a fox with a bushy tail in the style of Van Gogh`* However, it is evident that Van Gogh never painted `a fox with a bushy tail`, making this a *synthetic/fake* prompt. The generated outputs for these *synthetic/fake* prompts may fail to evoke sufficient signal to represent Van Gogh's style, which can reduce erasure effectiveness in output-based methods such as ESD, AP, AGE, and ACE.
> >
> > - **Effectiveness of Diversified Embedding Mixup in attention-based methods.** In methods like UCE, Diversified Embedding Mixup focuses on diversifying the embedding of `Van Gogh` directly, without introducing *synthetic/fake* prompts. This allows for more effective unlearning of the artistic style concept while maintaining preservation.

---

> ### Author Response · Authors · 2025-11-27
>
> Dear Reviewer **sj5a**,
>
> We would like to provide additional experiments on the suggested adversarial-training–based unlearning methods, including **AdvUnlearn**, **RECE**, and **Receler** (previously provided, but reproduced here for clarity). AdvUnlearn and Receler belong to the *output-based unlearning* family, following the ESD framework, while RECE is an *attention-based* method with a closed-form solution similar to UCE.
>
> We integrate our proposed diversification technique into these keyword-based frameworks, resulting in three diversified variants: **Diversified-AdvUnlearn**, **Diversified-RECE**, and **Diversified-Receler**. We evaluate them on two erasure tasks: removing the celebrity *Henry Cavill* and removing *ImageNette* objects, using the same experimental settings detailed in Appendix B.1.1. The code has also been updated in the anonymous repository for further inspection.
>
> From the first table, we observe that:
>
> 1. **Compared to Receler**, our diversified counterpart achieves *equally strong unlearning performance* (both methods reach zero GPT-score and very low CLIP-i), while achieving *substantially better preservation performance*, improving by **+8.53 GPT-score** or **+3.7 CLIP-i**.
> 2. **Compared to RECE**, our diversified variant achieves slightly better preservation performance, along with a noticeably improved unlearning performance.
>
> The second table shows that, relative to AdvUnlearn, our Diversified-AdvUnlearn variant achieves *comparable unlearning effectiveness* while providing a *much stronger preservation ability*, including a notable **+30% PSR-5 improvement** on the task of erasing five objects simultaneously.
>
> Taken together, these results—along with the extensive experiments on recent state-of-the-art methods such as **ACE (CVPR 2025)** and **AGE (ICLR 2025)**—further strengthen our claim regarding both the *effectiveness* and the *generality* of our diversified unlearning approach. Our method not only enhances unlearning quality but also consistently improves preservation across diverse keyword-based unlearning frameworks.
>
> | **Method**| **Erasure** || **Preservation** ||||
> |-|-|-|-|-|-|-|
> || CLIP-i↓    | GPT-score↓ | LPIPS↓| CLIP-i↑ | CLIP-t↑ | GPT-score↑|
> | **Receler**| **51.61**| **0.00**| 0.57| 73.93   | 25.27  | 47.07|
> | **Diversified-Receler**| 52.90| **0.00**| **0.52**| **77.65** | **26.17** | **55.60** (+8.53%) |
> | **RECE**| 59.42| 8.05| **0.79**| 77.34| 29.32| 77.42|
> | **Diversified-RECE**| **58.57**| **7.78**| **0.79**| **77.95**| **29.64**| **77.87**|
>
> Quantitative results of erasing `Henry Cavill' using baseline Receler and RECE compared with the diversified counterpart, as detailed in Appendix B.1.1.
>
> | **Object**| **'Garbage Truck' erasure**|||| **'Five objects' erasure**||||
> |-|-|-|-|-|-|-|-|-|
> || ESR-1| ESR-5|PSR-1|PSR-5|ESR-1|ESR-5|PSR-1|PSR-5|
> | **AdvUnlearn**|**1.000**|**0.992**|0.768|0.936|**1.000**|**0.990**|0.459|0.604|
> | **Diversified-AdvUnlearn**|0.996|0.976|**0.807**|**0.952**|0.988|0.974|**0.732**| **0.916** |
>
> Quantitative results of object erasure using keyword-based methods compared with their diversified counterparts, evaluated across prompts of varying complexity levels. Erasure Success Rate (ESR) and Preservation Success Rate (PSR), computed with a pre-trained ResNet-50, where higher values indicate better performance. Overall, under the setting of simultaneously erasing five objects.

---

> ### Author Response · Authors · 2025-11-27
>
> Dear Reviewer **sj5a**,
>
> Thank you once again for the time and care you invested in reviewing our submission. We sincerely appreciate your constructive feedback, which has helped us strengthen the paper. We are writing to kindly remind you that we have now posted a detailed response to your comments.
>
> In our rebuttal, we have specifically addressed the key concerns you raised:
>
> #### **1. Detailed analysis of the impact and design considerations across concept categories**
> In Section 4.5 and Appendix C.6, we provide an ablation study examining how the diversity and contextual complexity of contextualized prompts influence the performance of our method, with three key observations:
> (i) diversity is essential,
> (ii) moderate contextual complexity is most effective, and
> (iii) increasing the number of prompts yields diminishing returns.
> Although we did not conduct deeper category-specific analyses for individual concept groups, these observations generalize well across categories and consistently lead to strong empirical performance.
>
> #### **2. Experiments with more recent diffusion models**
> Because our approach is designed as an add-on module built on top of existing unlearning baselines, we follow the experimental settings established by state-of-the-art unlearning methods such as ACE (CVPR 2025) and AGE (ICLR 2025). Furthermore, our method is model-agnostic, and extending it to more recent diffusion models (e.g., SD 3.5, FLUX) is straightforward. We view this as promising future work that will enable broader and more comprehensive benchmarking.
>
> #### **3. Enhancing adversarial training–based unlearning methods**
> Our method functions as an **add-on module** that can be integrated into existing unlearning baselines. When combined with adversarial training–based approaches such as **Receler**, our method improves preservation while retaining the strong erasure performance inherent to these adversarial baselines. Similar improvements are also observed in our new experiments with **AdvUnlearn** and **RECE**.
>
> #### **4. The impact of the hyperparameter α in Diversified Embedding Mixup**
> Our experiments on varying α show that increasing the contextual influence injected into the target concept leads to a noticeable drop in preservation performance, while providing only a slight improvement in erasure effectiveness. Setting α = 0.999 offers a balanced trade-off between unlearning and preservation, and is therefore adopted throughout the paper.
>
> #### **5. GPT-score over identity similarity as an evaluation metric**
> In addition to commonly used metrics, we adopt GPT-score as an alternative to identity-similarity measures from face recognition models. This choice is motivated by the discrepancy between diffusion-generated images—produced by general-purpose diffusion models—and the real-image data used to train face recognition systems. Moreover, GPT-score offers greater generality, enabling evaluation not only for celebrity-related concepts but also for a broader range of semantic categories.
>
> ---
>
> We would be sincerely grateful if you could take a moment to review our detailed responses. Please feel free to reach out with any additional questions during the remaining days of the discussion period — we are happy to provide further clarification if needed.
>
> Thank you again for your valuable insights.
>
> **Best regards,**
> *The Authors of Submission 3912*

---

> ### Comment · Reviewer_sj5a · 2025-11-27
>
> Thanks for the reply and the comprehensive experiments. I still have a few remaining questions:
>
> 1. Figure 4 studies different levels of contextualized prompts. Since contextual prompts are core to the method, is there a principle or guideline for constructing them? For example, for “Henry Cavill,” would “man” or “woman” work better—and why? A clear principle would avoid random guessing.
>
> 2. In my previous review, I recommended two open-sourced works [1,2] for comparison. The authors instead chose Receler. Is there a specific reason for this decision?
>
> 3. It would be helpful to add a brief experiment or discussion on applying the method to newer diffusion models. I do not mean the main experiments must switch models—just a small validation or comment.
> 4. In Table 7, the proposed method appears to perform worse than the baselines, or only shows very marginal improvements. This seems inconsistent with the conclusions drawn from the other experiments. Could the authors clarify this?
>
> [1] Zhang Y, Chen X, Jia J, et al. Defensive unlearning with adversarial training for robust concept erasure in diffusion models.
>
> [2] Gong C, Chen K, Wei Z, et al. Reliable and efficient concept erasure of text-to-image diffusion models.

---

> > ### Author Response · Authors · 2025-12-02
> >
> > Dear Reviewer sJ5a,
> >
> > We thank the reviewer for your appreciation of the comprehensiveness of our rebuttal. We would like to address the remaining concerns as follows:
> >
> > **1. Figure 4 studies different levels of contextualized prompts. Since contextual prompts are core to the method, is there a principle or guideline for constructing them? For example, for “Henry Cavill,” would “man” or “woman” work better—and why? A clear principle would avoid random guessing.**
> >
> > **Answer:**
> >
> > In Table 4, we analyze different levels of contextualized prompts. The principles used to construct these prompts are described in Appendix B.1 for the five concept categories. Below, we provide a more concrete example for the target concept *celebrity* `Henry Cavill`:
> >
> > ### **B1.**
> > Starting from the target concept to be erased, `Henry Cavill`, the prompt `Henry Cavill` or `A photo of Henry Cavill` is used as the Level-0 prompt.
> >
> > ### **B2.**
> > We then define the construction rules for higher levels:
> >
> > - **Level-1:** `Henry Cavill` performing a simple action
> >   *e.g.,* `A photo of Henry Cavill gesturing`.
> >
> > - **Level-2:** introduces interactions with another entity
> >   *e.g.,* `A photo of Henry Cavill walking with a man`.
> >
> > - **Level-3:** further enriches the scene with two interacting entities
> >   *e.g.,* `A photo of Henry Cavill listening to a man beside a bookshelf`.
> >
> > - **Level-4:** includes more natural and semantically complex scenarios
> >   *e.g.,* `A photo of Henry Cavill jogging across a bridge at dawn beside a man wearing a campaign T-shirt`.
> >
> > ### **B3.**
> > Using an LLM (e.g., ChatGPT), we randomly generate diverse prompts for each level following the construction rules and examples defined in B2.
> >
> > ---
> >
> > Before designing the different levels of prompt complexity, we begin with the following assumption, which serves as the foundation of our Diversified Prompting strategy for output-based unlearning methods.
> >
> > ### **Assumption.**
> > For output-based methods such as ESD, AGE, and ACE, the strength and persistence of the target-concept signal in the denoising trajectory are crucial for effective unlearning. When a simple, low-context prompt is used (e.g., `A photo of Henry Cavill`), the target concept remains strongly represented across timesteps. In contrast, when the prompt is enriched with substantial contextual information — for example, `A photo of Henry Cavill jogging across a bridge at dawn beside a man wearing a campaign T-shirt` — the model must allocate its denoising capacity to render multiple competing cues, including the bridge, dawn lighting, jogging motion, the additional person, and clothing details.
> >
> > As the denoising process progresses, these competing contextual factors increasingly overshadow the identity-related signal of `Henry Cavill`. Consequently, the target-concept signal becomes diluted throughout the trajectory, weakening the effective unlearning signal and ultimately reducing the performance of the unlearning method.
> >
> > This assumption motivates our careful and progressively controlled design of prompt-complexity levels. The empirical results in Fig. 4 strongly support this hypothesis.

---

> > > ### Author Response · Authors · 2025-12-02
> > >
> > > **2. In my previous review, I recommended two open-sourced works [1,2] for comparison. The authors instead chose Receler. Is there a specific reason for this decision?**
> > >
> > > **Answer:**
> > >
> > > Dear Reviewer sj5a,
> > >
> > > We sincerely apologize for not being able to include the results of AdvUnlearn [1] and RECE [2] alongside those of Receler in the original submission. The preparation and reproduction of AdvUnlearn and RECE took significantly longer than expected due to their complex experimental pipelines and additional implementation dependencies. Nevertheless, we have now provided these results in a recent comment, and for completeness, we restate them below:
> > >
> > > | **Method**| **Erasure** || **Preservation** ||||
> > > |-|-|-|-|-|-|-|
> > > || CLIP-i↓    | GPT-score↓ | LPIPS↓| CLIP-i↑ | CLIP-t↑ | GPT-score↑|
> > > | **Receler**| **51.61**| **0.00**| 0.57| 73.93   | 25.27  | 47.07|
> > > | **Diversified-Receler**| 52.90| **0.00**| **0.52**| **77.65** | **26.17** | **55.60** (+8.53%) |
> > > | **RECE**| 59.42| 8.05| **0.79**| 77.34| 29.32| 77.42|
> > > | **Diversified-RECE**| **58.57**| **7.78**| **0.79**| **77.95**| **29.64**| **77.87**|
> > >
> > > Quantitative results of erasing `Henry Cavill' using baseline Receler and RECE compared with the diversified counterpart, as detailed in Appendix B.1.1.
> > >
> > > | **Object**| **'Garbage Truck' erasure**|||| **'Five objects' erasure**||||
> > > |-|-|-|-|-|-|-|-|-|
> > > || ESR-1| ESR-5|PSR-1|PSR-5|ESR-1|ESR-5|PSR-1|PSR-5|
> > > | **AdvUnlearn**|**1.000**|**0.992**|0.768|0.936|**1.000**|**0.990**|0.459|0.604|
> > > | **Diversified-AdvUnlearn**|0.996|0.976|**0.807**|**0.952**|0.988|0.974|**0.732**| **0.916** |
> > >
> > > Quantitative results of object erasure using keyword-based methods compared with their diversified counterparts, evaluated across prompts of varying complexity levels. Erasure Success Rate (ESR) and Preservation Success Rate (PSR), computed with a pre-trained ResNet-50, where higher values indicate better performance. Overall, under the setting of simultaneously erasing five objects.
> > >
> > > We hope that the inclusion of these results helps clarify the comparative performance and fully addresses your concerns.

---

> ### Author Response · Authors · 2025-12-02
>
> **3. It would be helpful to add a brief experiment or discussion on applying the method to newer diffusion models. I do not mean the main experiments must switch models—just a small validation or comment.**
>
> **Answer:**
>
> We thank reviewer sj5a for the valuable feedback, and we agree that discussing the applicability of our method to newer diffusion models would be beneficial. In essence, Diversified Unlearning is a distributional "add-on" framework that is not tied to any specific architecture: diversified prompting modifies only the input distribution through natural-context prompts for the concept to be erased, while diversified embedding mixup operates in the embedding space of the text encoder. Therefore, both components can be applied to different families of Text-to-Image models without modifying their core architecture or training pipeline.
>
> For newer variants of Stable Diffusion such as 2.0 and 2.1, although they are still built on the Latent Diffusion framework with a 2D U-Net and cross-attention blocks in the down/mid/up branches, they introduce several important differences compared to SD v1.4: (i) The text encoder is replaced by OpenCLIP ViT-H/14, resulting in a higher-dimensional embedding space; (ii) The training objective switches from the default noise prediction (v1.x) to v-prediction; (iii) The default image resolution may be increased to 768 instead of 512; (iv) The VAE/autoencoder and noise schedule are updated. With these changes, our two approaches still apply directly as follows: for diversified prompting in output-based methods (e.g., ESD, AP, AGE), one only needs to replace the noise-prediction loss with its v-prediction counterpart. For diversified embedding mixup in Eq. (6), we replace $\tau$ with the SD2 text encoder, perform mixup at the token-embedding level for tokens corresponding to the erased concept, and optimize the cross-attention projections that process the textual conditions. The preservation and anchor-based components remain unchanged, except for larger embedding dimensions and a different tokenizer.
>
> For the FLUX architecture, it adopts a multimodal DiT-style Transformer (MM-DiT), where text tokens and image tokens interact through multimodal self-attention rather than the U-Net cross-attention mechanism, with an encoder-only text encoder. In this setting, diversified prompting can still be directly applied because it only alters the prompt distribution, similar to SD2 when switching from noise prediction to v-prediction. For diversified embedding mixup, we can also apply mixup at the token-embedding level and optimize the projection/adapter parameters responsible for text→image interactions within MM-DiT blocks. Since mixup operates on token embeddings and our objective concerns contextual distribution, changes in the tokenizer/encoder of FLUX do not alter the fundamental nature of our method.
>
> Due to time constraints during the rebuttal process, we were not able to complete experiments and evaluations on newer versions of Stable Diffusion and FLUX. Once again, we thank the reviewer for the insightful comments.
>
> **4. In Table 7, the proposed method appears to perform worse than the baselines, or only shows very marginal improvements. This seems inconsistent with the conclusions drawn from the other experiments. Could the authors clarify this?**
>
> **Answer**
>
> **Single-object erasure.**  For objects such as *`Cassette Player`* and *`Garbage Truck`*, baseline methods including UCE, ESD, AP, AGE, and ACE already achieve near-perfect erasure performance. Consequently, our method remains comparable in the erasure metric.
>
> **Five-object joint erasure.**  When erasing five objects simultaneously, the performance of all baselines noticeably drops, creating more room to observe the benefits of our Diversified Unlearning strategy. In this setting, our method consistently improves the erasure performance across all baselines, with significant gains in ESR-5: Diversified-AGE (**+30.9%**), Diversified-AP (**+6.7%**), and Diversified-ACE (**+5.9%**).
>
> **Explanation for object category.**  Unlike living entities such as celebrities or copyrighted characters, objects (*`Cassette Player`*, *`Church`*, *`Garbage Truck`*, *`Parachute`*, *`French Horn`*) are non-living and do not naturally afford simple action-based prompts (e.g., `A photo of Henry Cavill gesturing`). As a result, object prompts must inherently place the target object within a specific scene. These contextual elements reduce the emphasis on the target concept. Combined with the fact that single-object erasure baselines already perform nearly perfectly, the benefits of Diversified Unlearning become clearly visible only in the more challenging multi-object setting, where there is significantly more headroom for improvement.

---

### Official Review · Reviewer_fJw7 · 2025-10-31

**Soundness:** 2
**Presentation:** 3
**Contribution:** 2
**Rating:** 2
**Confidence:** 5

**Summary:**

In this paper, the authors propose a data augmentation method to produce more diverse samples, helping erase concepts from text-to-image diffusion models.

**Strengths:**

1. The paper is well-written, easy for readers to follow.
2. The proposed method is intuitive, targeting the issue.

**Weaknesses:**

1. Require for more verifying experiments.

a) First, one immediately approach is to use prompts to construct pairs of data. For example, <p1, p2> where p1 is a prompt with the word nudity and p2 is the same prompt but without the word nudity. Then we use the pairwise data to train the model. In this scenaro, what is the advantage of this proposed method?

b) How do we evaluate the precision of the sum operation directly on the embedding vectors. As we know, the latent space is complex and the embedding vector sum may destroy the original semantic.

c) If the embedding vector sum destroys the original semantic, it indicates that the semantic in the prompts is not important in this task. Can we directly use embedding vectors generated randomly as $c$?

d) In Eq5, why do we need to prepare $c_p$ additionally for preservation rather than using $c$ directly?

2. The baselines are old. Please compare your methods with more recent methods, such as RECE, Receler and AdvUnlearn.

**Questions:**

See weaknesses.
I think that there are many concerns to be addressed. More experiments and discussions are needed to evaluate the proposed method. If the authors can give convincing evidence, I will raise my rating.

---

> ### Author Response · Authors · 2025-11-22
>
> We thank the reviewer for providing the opportunity to discuss these points and for adjusting the score. We would like to address your remaining concerns as follows:
>
> **W1-a First, one immediately approach is to use prompts to construct pairs of data. For example, <p1, p2> where p1 is a prompt with the word nudity and p2 is the same prompt but without the word nudity. Then we use the pairwise data to train the model. In this scenario, what is the advantage of this proposed method?**
>
> **Answer:**
> As discussed in Section 2 (Background), a typical unlearning objective consists of two components: the *erasure loss* ($L_1$) and the *preservation loss* ($L_2$), each serving distinct purposes. Our **Diversified Prompting** formulation goes beyond conventional data augmentation; instead, it can be viewed as a mechanism to induce **contrastive tension** within the erasure loss through carefully designed prompt-based fine-tuning.
>
> Assume we aim to forget the target concept, *“Henry Cavill”*. In keyword-based approaches, given the input prompt *“Henry Cavill”* and its mapping to an anchor concept $c_a$ (e.g., “a man”, “a person”, or an empty neutral string “ ”), several issues arise:
>
> - First, Diversified Prompting applies only to **output-based methods** such as ESD, AP, and AGE—where the model primarily observes generated outputs. A synthesized image may contain peripheral details unrelated to the target concept. When performing standard unlearning, the model could inadvertently map both *“Henry Cavill”* and these additional contextual details to a neutral anchor concept $c_a$, thereby degrading the model’s preservation capability.
>
> - Moreover, the prompt *“Henry Cavill”* is encoded by the text encoder into a single fixed embedding that remains constant across unlearning iterations. In contrast, during the original diffusion model training, multiple image–prompt pairs place *“Henry Cavill”* in diverse contexts, leading to context-dependent variations in the text embedding after the encoder. This discrepancy limits the effectiveness of erasure when only a static embedding is used.
>
> The key innovation of the proposed **Diversified Prompting** framework lies in introducing **contrast within the erasure loss**:
>
> - By augmenting the input prompt with contextual information—forming pairs of {“context” + “target concept”} mapped to {“context” + “neutral anchor concept”} ($L_1$ in Eq. 5)—the model is explicitly encouraged to remove only the “target concept” while preserving the given context. This contextual contrast sharpens the erasure loss, compelling the model to focus its forgetting process precisely on the intended concept, thereby improving preservation inherently within $L_1$.
>
> - Furthermore, the prompt {“context” + *“Henry Cavill”*} produces **context-sensitive embeddings** of *“Henry Cavill”* through the interaction between the “target concept” and its surrounding context within the text encoder. This diversification in the embedding space enhances the representation variability of the “target concept”, leading to more robust and effective unlearning.
>
> When constructing training pairs $\langle p_1, p_2 \rangle$, where $p_1$ is a prompt containing the “target concept” and $p_2$ is the same prompt with the target concept removed, the process is not as straightforward as it may appear:
>
> - First, the semantic complexity of the selected prompts must be carefully controlled; choosing prompts that are overly complex can degrade the erasure performance.
>
> - Secondly, naively removing the “target concept” may distort the semantic integrity of $p_2$.
>   For example, mapping *“A photo of Henry Cavill waving”* to *“A photo of waving”* results in an unnatural and semantically broken prompt.

---

> ### Author Response · Authors · 2025-11-22
>
> **W1-b) How do we evaluate the precision of the sum operation directly on the embedding vectors. As we know, the latent space is complex and the embedding vector sum may destroy the original semantic.**
>
> **Answer:**
> As introduced in Section 3 (**Diversified Embedding Mixup**), we construct diversified representations of the target concept by performing a weighted summation between the embeddings of the *“target concept”* and its surrounding *“context”*. Specifically, in Equation 7 the mixed embedding is interpolated as:
>
> $$
> f(\tau(c_e), \mathbf{C})^i = \sum_{c \in \mathbf{C}} \frac{1}{|\mathbf{C}|}
> \left( \alpha \, \tau(c_e)^i + (1 - \alpha) \, \tau(c)^i \right)
> $$
>
> Where $\alpha \in [0,1]$ controls the relative contribution of the target and contextual semantics. This operation effectively produces a “diversified target concept” embedding, enabling richer and more robust representations than the conventional keyword-based approach.
>
> Before applying this strategy to the unlearning process, we conducted a pilot experiment to validate the semantic stability of such perturbations, as shown in Table 14. In this setup, we added Gaussian noise to the embedding of the prompt *“a photo of Cassette Player”*, and evaluated the model’s response using the DS-5 metric, which indicates the top-5 accuracy of concept detection.
>
> The results demonstrate that, with moderate noise levels (1 − α) ranging from 0.0 to 0.3, the model still generates the *“Cassette Player”* concept with high probability. This observation suggests that introducing controlled noise or contextual blending does not disrupt the semantic integrity of the target concept, while simultaneously promoting greater diversity in its embedding space.
>
> **W1-c) If the embedding vector sum destroys the original semantic, it indicates that the semantic in the prompts is not important in this task. Can we directly use embedding vectors generated randomly as c?**
>
> **Answer:**
> We thank the reviewer for this insightful suggestion. Following this idea, we implement a new variant of our method called *Noise-based Embedding Mixup*, which replaces the embedding of contextualized prompts with a random vector $n \sim \mathcal{N}(0, I)$. Specifically, the mixing function is defined as:
>
> $f(\tau(c_e), n)^i = \alpha \, \tau(c_e)^i + (1 - \alpha) \, n$ if token $i$ belongs to $c_e$.
>
> We conduct an additional experiment on the celebrity erasure task to compare our (Contextualized) Diversified Embedding Mixup with this new variant, and report the results in the table below.
>
> Interestingly, Noise-based Embedding Mixup (with weighting factor $1 - \alpha = 0.001$) achieves slightly lower unlearning performance compared to Diversified Embedding Mixup (1.5 points lower on CLIP-I and comparable GPT-score). However, it demonstrates considerably better preservation performance (2.5 points higher on CLIP-I and 12.5 points higher on GPT-score).
>
> This improvement in preservation performance can be explained intuitively as follows:
>
> 1. The mixed embedding using Noise-based Embedding Mixup $f(\tau(c_e), n)^i$ identifies the target concept less precisely than the Contextualized embedding $f(\tau(c_e), C)^i$, resulting in slightly lower unlearning performance.
>
> 2. Contextualized embeddings utilize embeddings of concepts related to the target concept $c_e$ as anchors. These related concepts are essentially keywords that have become entangled with other concepts (having similar entanglement problem as the target $c_e$). Consequently, even after mixing, the Contextualized embedding $f(\tau(c_e), C)^i$ still inherits these unintended relationships.
>
> 3. In contrast, the Noise-based Embedding $f(\tau(c_e), n)^i$ uses a random vector $n$ that is completely disentangled from any concepts. This produces a cleaner mixed embedding with minimal inherited entanglements, thereby yielding better preservation performance.
>
> We again thank the reviewer for this valuable suggestion and plan to investigate this direction further in future work.
>
> | **Method**| **Erasure**| | **Preservation** ||||
> |-|-|-|--|-|-|-|
> |  | CLIP-i↓| GPT-score↓| LPIPS↓| CLIP-i↑ | CLIP-t↑| GPT-score↑|
> | UCE| 60.59| 4.48| 0.65| 62.90| 23.97| 31.42|
> | **Diversified-UCE mixup**| **58.55**| **2.83**| 0.64| 66.92| 24.57| 24.40|
> | Noise-based UCE mixup    | 60.09| 2.85| **0.62**| **69.40** | **25.30** | **36.95**   |
>
> As can be seen from table, we compare unlearning and retention performance between our approach and the noise-based mixup method using the same weighting factor $\alpha$ to control the magnitude of the mixed embeddings. Specifically, our experiments were conducted on the task of erasing 10 celebrities. Removal performance was evaluated using 2,000 prompts across multiple complexity levels, and retention performance was assessed using 1,500 keyword-based prompts covering 15 other celebrities. Full experimental settings are provided in Section 4.1 and Appendix B.1.1.

---

> ### Author Response · Authors · 2025-11-22
>
> **W1-d) In Eq5, why do we need to prepare additionally for preservation rather than using c directly?**
>
> **Answer:**
> In our method, we primarily focus on the *erasure* component, while the *preservation* part is kept consistent with that of the baseline methods. Specifically, the ESD baseline does not include a preservation term $C_p$, whereas methods such as UCE, AP, and AGE incorporate this component.
>
> To ensure a fair and *comparable* evaluation, we include $C_p$ only when it is also present in the corresponding baseline.
>
> **W2. The baselines are old. Please compare your methods with more recent methods, such as RECE, Receler and AdvUnlearn.**
>
> **Answer:**
> We respectfully note that the baselines employed in our experiments are not outdated; rather, they represent recently introduced methods: AP (NeurIPS 2024), AGE (ICLR 2025), and ACE (CVPR 2025).
> Furthermore, our approach functions as an **add-on module** rather than a standalone unlearning framework, and is therefore not directly comparable to independent methods such as RECE or AdvUnlearn.
>
> Through comprehensive experiments, we have demonstrated that our proposed **Diversified Unlearning** method consistently enhances performance when applied on top of these baseline approaches.
>
> Following the reviewer's suggestion, we conducted an additional experiment with Receler, an adversarial training-based unlearning approach under experiment of erasing 'Henry Cavill' as detailed in Appendix B.1.1, and provide the results in the table below.
>
> | **Method**                | **Erasure** |      | **Preservation** |        |        |                   |
> |----------------------------|------------|------|-----------------|--------|--------|------------------|
> |                            | CLIP-i↓    | GPT-score↓ | LPIPS↓         | CLIP-i↑ | CLIP-t↑ | GPT-score↑       |
> | Receler                    | **51.61**  | **0.00**   | 0.57           | 73.93   | 25.27  | 47.07           |
> | **Diversified-Receler**    | 52.90      | **0.00**   | **0.52**       | **77.65** | **26.17** | **55.60** (+8.53%) |
>
> As shown in the table above, integrating Diversified Unlearning into Receler preserves the original model’s erasure performance while markedly enhancing preservation, yielding an 8.53% improvement in the GPT-score. Beyond the quantitative metrics, we also provide qualitative examples in the anonymous GitHub repository to enable direct visual comparison between Receler and Diversified-Receler. Although additional fine-tuning of our add-on module could potentially lead to even stronger performance, we intentionally employ the same default configuration used in our other experiments to ensure fairness and consistency.
>
> These additional results—together with the comprehensive experiments conducted on recent state-of-the-art unlearning methods such as *ACE* (CVPR 2025) and *AGE* (ICLR 2025)—further reinforce both the effectiveness and the broad applicability of our proposed approach.

---

> > ### Comment · Reviewer_fJw7 · 2025-11-26
> > **Response to Rebuttal**
> >
> > Thank you very much for your efforts to address my concerns. I believe your reply has resolved most of my theoretical questions. Nevertheless, I remain skeptical about the effectiveness of Eq7.
> >
> > 1. In Tab.14, a straightforward concept, "Cassette Player", is evaluated. It has a clear semantic meaning. More abstract concepts are needed. For example, nudity, objects (church), and painting styles. For these concepts that lack a clear direction, applying this method with various $\alpha$ may lead to inaccuracies.
> >
> > 2. The default $\alpha$ is a large value 0.999. It indicated that a tiny perturbation was added to the embeddings. Under this setting, it is natural to maintain the generation of concepts. I think this scenario is equal to adversarial learning. In your experiment, GPT-scores of Diversified-UCE and Noise-based UCE are close, as I had guessed. In addition, noise does not need to be N(0, I). It should be consistent with $\tau (c)$ from the perspective of the mean and variance.
> >
> > 3. From your experiments in Tab.2, the effectiveness of your method is not stable. Your method cannot stably enhance existing methods, especially for preservation metrics.
> >
> > 4. An additional suggestion: You can distinguish your methods for output-based unlearning and attention-based unlearning in the method section by plotting a figure and in the experiment section by labelling method categories.
> >
> > Thank your rebuttal again. While many concerns have been addressed, I will keep my rating because of the above reasons. I have also noticed that some reviewers gave positive feedback. I present these differing perspectives to ACs to make a decision.

---

> ### Author Response · Authors · 2025-11-27
>
> Dear Reviewer fJw7,
>
> We thank the reviewer for taking the time to read our rebuttal and for providing further feedback. We are pleased that most of the theoretical concerns have been clarified. For the remaining points, we would like to offer additional clarification as follows.
>
> We would also like to highlight that we have included new experimental results on three adversarial-training–based unlearning methods—AdvUnlearn, RECE, and Receler—which further reinforce the effectiveness and generality of our proposed approach.
>
> We hope that our detailed rebuttal successfully addresses your remaining concerns. Should you have any additional questions, we would be more than happy to provide further clarification.
>
> **1. Behavior in different concepts when injecting noise.**
>
> **Answer:**
>
> We follow the reviewer’s suggestion and conduct additional experiments to examine how latent representations for specific concepts (e.g., *nudity*, *church*) behave under injected noise. We conducted experiments by generating 100 images for each setting under noise injection for both the *nudity* and *church* concepts. For the *church* concept, we use a ResNet-based classifier to detect whether the concept appears in the generated image and report its detection score. For the *nudity* concept, we use NudeNet to detect exposed body parts in the generated images and report the total number of detected parts. From both tables, we observe that the model remains capable of generating the target concept even with substantially distorted text embeddings (with $\alpha = 0.3$), consistent with our findings for the “Cassette Player’’ concept. These results suggest that the phenomenon is robust across concepts with different degrees of abstraction.
>
> This behavior can be explained by noting that the text embedding of the prompt functions primarily as a conditioning signal that guides the diffusion model during denoising. While injecting noise perturbs this guidance, the modified embedding still preserves a significant amount of the original concept information. Additionally, because the noisy embedding remains highly similar to the original—due to the adjustment being controlled by the stable coefficient $\alpha$—the resulting embedding continues to reliably represent the intended concept.
>
> ---
>
> ### **Table: Church Concept Under Noise Injection**
>
> | **SD 1.4** | α = 0 | α = 0.1 | α = 0.3 | α = 0.5 | α = 0.8 | α = 1.0 |
> |-|-|-|-|-|-|-|
> | **DS-1**  | 86| 80| 75| 38| 1| 1|
> | **DS-5**  | 100| 100| 91| 54| 4| 2|
>
> ---
>
> ### **Table: Nudity Concept Under Noise Injection**
>
> | **SD 1.4**| α = 0 | α = 0.1 | α = 0.3 | α = 0.5 | α = 0.8 | α = 1.0 |
> |-|-|-|-|-|-|-|
> | **Num of unsafe images**| 100| 100| 91| 48| 4| 2|
> | **Num of exposed body parts**| 779| 779| 505| 174| 10| 2|

---

> ### Author Response · Authors · 2025-11-27
>
> **2. Why the value of $\alpha$ and the behavior of the text embedding with noise.**
>
> **Answer:**
>
> We agree that the default large value of $\alpha$ may initially seem surprising. While the mixed embeddings may still appear visually similar to the original concept from a human perspective, the behavior is different from the perspective of the text encoder. In practice, this large $\alpha$ provides a favorable trade-off: it preserves sufficient information for the encoder to reliably identify the target concept while still introducing enough diversity to benefit the unlearning process.
>
> Mathematically, consider the mixed embedding:
>
> $$
> y = \alpha x + (1 - \alpha) z, \quad x, y \in \mathbb{R}^d,
> $$
>
> where $z$ is either the Diversified Token Embedding or the Noise-based Embedding.
> The signal–to–noise ratio (SNR) scales proportionally to:
>
> $$
> \frac{\alpha^2 \||x\||^2}{(1 - \alpha)^2 d},
> $$
>
> and the cosine similarity between $x$ and $y$ approximates:
>
> $$
> \frac{\alpha \||x\||}{\sqrt{\alpha^2 \||x\||^2 + (1 - \alpha)^2 d}}.
> $$
>
> These expressions illustrate that in very high-dimensional spaces (large $d$), keeping $(1 - \alpha)$ small is crucial to maintaining both a high SNR and strong alignment between $x$ and $y$. This explains why a numerically large $\alpha$ remains appropriate despite appearances.
>
> Second, the performance of Diversified-UCE and Noise-based UCE does not contradict our conclusion regarding the limitations of keyword-based unlearning or the importance of incorporating contextual diversity. Both variants share the same underlying principle—injecting diversity into the target embedding—while differing in how that diversity is generated. Noise-based UCE introduces random perturbations near the target embedding $\tau(c_e)$, whereas Diversified-UCE provides a more directed and semantically meaningful diversification by leveraging contextualized embeddings.
>
> Finally, to further support the effectiveness of our Diversified Token Mixup strategy, we conducted additional experiments on RECE—an adversarial-training–based method with a closed-form update similar to UCE—alongside AdvUnlearn and Receler, two robust output-based unlearning frameworks built on ESD. These results consistently reinforce our claims regarding the effectiveness and generality of our diversified unlearning approach.
>
> Specificially, we integrate our proposed diversification technique into these keyword-based frameworks, resulting in three diversified variants: **Diversified-AdvUnlearn**, **Diversified-RECE**, and **Diversified-Receler**. We evaluate them on two erasure tasks: removing the celebrity *Henry Cavill* and removing *ImageNette* objects, using the same experimental settings detailed in Appendix B.1.1. The code has also been updated in the anonymous repository for further inspection.
>
> From the first table, we observe that:
>
> 1. **Compared to Receler**, our diversified counterpart achieves *equally strong unlearning performance* (both methods reach zero GPT-score and very low CLIP-i), while achieving *substantially better preservation performance*, improving by **+8.53 GPT-score** or **+3.7 CLIP-i**.
> 2. **Compared to RECE**, our diversified variant achieves slightly better preservation performance, along with a noticeably improved unlearning performance.
>
> The second table shows that, relative to AdvUnlearn, our Diversified-AdvUnlearn variant achieves *comparable unlearning effectiveness* while providing a *much stronger preservation ability*, including a notable **+30% PSR-5 improvement** on the task of erasing five objects simultaneously.
>
> ### **Table: 'Henry Cavill' Concept Under RECE and Receler compared with their Diversified methods.**
>
> | **Method**| **Erasure** || **Preservation** ||||
> |-|-|-|-|-|-|-|
> || CLIP-i↓    | GPT-score↓ | LPIPS↓| CLIP-i↑ | CLIP-t↑ | GPT-score↑|
> | **Receler**| **51.61**| **0.00**| 0.57| 73.93   | 25.27  | 47.07|
> | **Diversified-Receler**| 52.90| **0.00**| **0.52**| **77.65** | **26.17** | **55.60** (+8.53%) |
> | **RECE**| 59.42| 8.05| **0.79**| 77.34| 29.32| 77.42|
> | **Diversified-RECE**| **58.57**| **7.78**| **0.79**| **77.95**| **29.64**| **77.87**|
>
> ### **Table: Imagenette Concept Under AdvUnlearn and its Diversified variant.**
>
> | **Object**| **'Garbage Truck' erasure**|||| **'Five objects' erasure**||||
> |-|-|-|-|-|-|-|-|-|
> || ESR-1| ESR-5|PSR-1|PSR-5|ESR-1|ESR-5|PSR-1|PSR-5|
> | **AdvUnlearn**|**1.000**|**0.992**|0.768|0.936|**1.000**|**0.990**|0.459|0.604|
> | **Diversified-AdvUnlearn**|0.996|0.976|**0.807**|**0.952**|0.988|0.974|**0.732**| **0.916** |

---

> > ### Author Response · Authors · 2025-11-27
> >
> > **3. From your experiments in Tab.2, the effectiveness of your method is not stable. Your method cannot stably enhance existing methods, especially for preservation metrics.**
> >
> > **Answer:**
> >
> > Regarding the nudity erasure task specifically, Table 2 presents the erasure and preservation performance when applying Diversified Unlearning on top of the baseline methods. While erasure performance improves for baselines such as UCE, ESD, AP, and AGE, ACE exhibits a slight decline. However, examining resilience against recovery attacks using adversarial prompts from Ring-A-Bell, as illustrated in Table 11, we observe that **Diversified-ACE maintains the original ACE model's strong erasure capability** (near 0% Attack Success Rate).
> >
> > We acknowledge that preservation performance shows a slight drop; importantly, this outcome is *expected and inherent to the task*. Nudity prompts in I2P and Ring-A-Bell are explicitly designed to elicit highly entangled visual elements (e.g., skin exposure, pose, body structure), many of which overlap with the contextual cues required for preservation. As a result, removing nudity often necessitates suppressing visual patterns that are difficult to disentangle from the surrounding context, making perfect preservation fundamentally more challenging for all methods, including ours.
> >
> > From a broader perspective, our method consistently leads to preservation improvements across all baselines on **celebrities (Table 1)**, **artistic styles (Table 10)**, and even yields **modest gains on copyrighted characters (Table 1)**.
> >
> > In summary, while nudity erasure inherently involves stronger concept entanglement that limits preservation, our method demonstrates clear preservation gains across most tasks and provides substantial improvements in targeted removal and robustness — the central objectives of concept unlearning.
> >
> > **4. An additional suggestion: You can distinguish your methods for output-based unlearning and attention-based unlearning in the method section by plotting a figure and in the experiment section by labelling method categories.**
> >
> > **Answer:**
> >
> > We thank the reviewer for the helpful suggestion. In the revised version, we will provide a clearer distinction between the baseline methods, explicitly indicating which ones fall under output-based approaches and which ones belong to attention-based approaches.

---

### Author Response · Authors · 2025-12-02
**Global Response**

We appreciate the Area Chair and reviewers for their effort on reviewing our paper and their insightful feedback. In this global response, we would like to summarize key contributions made in our work and additional supportive results that we conduct in the rebuttal to address the reviewers' remaining concerns.

**Contributions**

Most previous concept unlearning methods are typically formulated in a *keyword-based* manner: a concept is represented by one or a few textual tokens. However, visual concepts are inherently multi-dimensional and can normally be described in numerous textual forms, therefore, relying on keywords to specify the target concept leads to two main issues: (1) vulnerability against recovery attacks and (2) inadvertently damaging related-benign concepts because of the over general representation of the target concept (via few keywords).

Our paper proposes **Diversified Unlearning**, a distributional framework that generalizes the way to identify a concept, by considering with diversified context. Although looks intuitively simple, our method can effectively the two aforementioned issues:
(1) the target concept has been much richer. Therefore, our method can be much robust against recovery attacks even with recent SOTA ones as demonstrated in Section 4.4;
(2) by composing the target concept with specific benign contexts and designing a loss function that emphasizes distinguishing the concept to be removed from the benign contexts that must be preserved, we mitigate the impact of the unlearning process on related-benign concepts, as demonstrated in our experiments in Sections 4.1, 4.2, and 4.3.

Our contributions have been acknowledged by the reviewers. More specifically, the reviewers consistently commend the paper for its clarity, strong motivation, and intuitive methodology (FJW7). They highlight the novelty of shifting from keyword-based unlearning to diversification-based concept unlearning, which offers a fresh and promising direction (SJ5A). The proposed approach is simple, plug-and-play, and effectively boosts the performance of existing unlearning methods (SJ5A, WWWB). The experiments on diffusion models are extensive and compelling, demonstrating superior results across five concept-erasure categories (SJ5A). The paper also provides thorough analyses of prompt diversity and contextual complexity, supported by clear examples that make the method easy to understand (WWWB). Overall, the reviewers find the work well-written, practical, and impactful for advancing more robust and trustworthy generative AI systems.

**Additional supportive results**

We are also grateful for the insightful questions that helped us further improve the paper. First, since our method is an add-on approach designed to enhance state-of-the-art baselines such as AP, AGE, and ACE, we address the reviewers’ concerns regarding its effectiveness when applied to robustness-oriented adversarial unlearning methods such as Receler, AdvUnlearn and RECE. The results show that our diversified variants yields notable gains in preservation quality while maintaining competitive unlearning performance over all adversarial unlearning methods.

Second, for the Diversified Embedding Mixup method, we provide a more complete evaluation of the hyperparameter $\alpha$, showing that increasing contextual influence strengthens erasure but comes at the cost of preservation. This is consistent with the observations reported in Table 14 and indicates that injecting a small amount of contextual influence into the latent space helps increase the diversity of the target concept. However, the original semantics may be distorted when the contextual influence becomes stronger, causing preservation performance to fall below the baseline. These results further highlight the necessity of Diversified Unlearning over keyword-based unlearning, as our approach effectively addresses the limitations of representing target concepts and yields stable improvements across all baselines.

While it is unfortunate that we do not have the opportunity to further discuss these points with the reviewers, we believe that our comprehensive responses have addressed their remaining concerns and have further strengthened the contributions and validity of our diversified unlearning method.

---

### Meta-Review · Area_Chair_WyfA · 2026-01-07

**Summary:**

The paper proposes a diversified formulation of concept unlearning for text-to-image diffusion models, motivated by the observation that concepts are multi-dimensional and cannot be reliably represented by single keywords. While reviewers generally found the idea intuitive and the paper clearly written, several concerns significantly influenced the final recommendation.

The main issues raised relate to the robustness and consistency of the empirical gains, particularly across different concept categories and unlearning baselines. In several settings, improvements in preservation or erasure were inconsistent or marginal, and performance degraded for certain important cases such as artistic styles or highly entangled concepts. Reviewers also expressed reservations about whether the proposed embedding mixup mechanism provides a principled improvement over keyword-based unlearning, or whether its effectiveness depends on small perturbations and careful tuning.

In addition, although the authors provided extensive experiments, the evaluation remains limited to a single diffusion backbone, which weakens claims of general applicability. Importantly, despite detailed rebuttals, multiple reviewers, especially fJw7, maintained negative reviews, indicating a lack of consensus on whether this work meets ICLR's acceptance criteria.

**Reviewer Concerns:**

Addressed Concerns:
- Reviewer fJw7: Theoretical questions (advantage over prompt-pair methods, embedding sum validity) resolved with experimental evidence; abstract concept (nudity/church) experiments added; mathematical justification for α=0.999 provided; comparisons with AdvUnlearn/RECE/Receler included.
- Reviewer sj5a: Prompt construction principles (leveled design for concepts like Henry Cavill) clarified; Table 7 inconsistency explained (single-object baselines near-perfect, gains visible in multi-object tasks); α ablation study conducted; discussion on applicability to newer models supplemented; comparisons with AdvUnlearn/RECE added.
- Reviewer WWWB: Hyperparameter α sensitivity analyzed; prompt complexity/quantity impact explained (moderate complexity optimal); recovery attack details (prompt levels/quantity) provided; direct comparison between hard/soft prompts added (diversified hard prompts outperform soft prompts in preservation).
- Reviewer jidj: Prompt quantity rules defined; LLM hallucination mitigated via verification pipelines; Receler comparison added; unrelated concept preservation mechanism clarified; safety validation (NudeNet for nudity detection) detailed.

Outstanding Concerns:
- Reviewer fJw7: Remains skeptical about Eq.7’s effectiveness for abstract concepts; questions why preservation improvements are unstable across baselines; no resolution on aligning noise distribution with τ(c) for α=0.999.
- Reviewer sj5a: Requests small-scale validation (not just discussion) on newer models (SD 3.5/FLUX) to confirm generalization; no further clarification on rare baseline underperformance.
- Reviewer jidj: Maintains doubts about the practicality of scaling prompt generation across diverse tasks.

**Reviewer Scores:**

- Reviewer fJw7: Initial score: 2. After rebuttal, the reviewer explicitly stated they would keep their rating, despite acknowledging that some theoretical questions were resolved.
- Reviewer sj5a: Initial score: 6. After rebuttal, concerns were reduced but not eliminated. Expected change: small positive or no change.
- Other reviewers likely to maintain their original scores.

---

### Decision · Program_Chairs · 2026-01-26

Reject